# Pulses of ocean acidification at the Triassic–Jurassic boundary

Molly Trudgill [1,2] ✉, James W. B. Rae [1], Ross Whiteford [1], Markus Adloff[3,4], Jessica Crumpton-Banks [1], Michael Van Mourik[1], Andrea Burke [1], Marieke Cuperus[5], Frank Corsetti[6], Daniel Doherty[4], William Gray [1,2], Rosanna Greenop[1], Wei-Li Hong [7,8], Aivo Lepland[7], Andrew McIntyre [1,9], Noor Neiroukh[1], Catherine V. Rose[1], Micha Ruhl [10], David Saunders[1], Magali M.F.R. Siri [5], Robert C. J. Steele[1], Eva E. Stüeken [1], A. Joshua West [6], Martin Ziegler [5] & Sarah E. Greene [4]

Mass extinctions have repeatedly perturbed the history of life, but their causes are often elusive. Ocean acidification has been implicated during Triassic–Jurassic environmental perturbations, but this interval lacks direct reconstructions of ocean pH. Here, we present boron isotope data from well-preserved fossil oysters, which provide evidence for acidification of ≥ 0.29 pH units coincident with a 2 ‰ negative carbon isotope excursion (the "main" CIE) following the end–Triassic extinction. These results suggest a prolonged interval of $CO_2$-driven environmental perturbation that may have delayed ecosystem recovery. Earth system modelling with cGENIE paired with our pH constraints demonstrates this was driven by predominantly mantle-derived carbon. Ocean acidification therefore appears to be associated with three of the five largest extinction events in Earth history, highlighting the catastrophic ecological impact of major perturbations to the carbon cycle in Earth's past, and possibly Earth's anthropogenically perturbed future.

Ocean acidification has been invoked as a kill mechanism in several mass extinctions in Earth's past, but establishing direct evidence of a linkage between pH and biodiversity in the geologic record is challenging. Much of our understanding of carbon cycle perturbations in the geological record is derived from stable carbon isotope ($\delta^{13}C$) excursions, but depending on the source of carbon, these excursions can represent different magnitudes of carbon input and associated ocean pH change[1]. Furthermore, while changes in ocean pH are important, the most salient feature of ocean acidification for organisms is the change to the saturation state of calcite and aragonite[2]. On long timescales, negative feedbacks (such as weathering and calcium

carbonate dissolution) buffer changes to the ocean saturation state[2]. Thus, appreciable carbonate undersaturation can only occur when the rate of carbon release and associated pH change is fast enough to outpace these restorative processes. Nonetheless, prolonged carbon emissions and associated acidification and warming may lead to other environmental impacts, including heat stress and deoxygenation, with the potential to impact the recovery from extinction events during intervals of prolonged environmental perturbation.

The end–Triassic mass extinction (~ 201 Ma)[3] represents one of the 5 largest mass extinctions of the Phanerozoic. A key driver for the extinction is thought to have been the emplacement of the Central

[1]School of Earth and Environmental Sciences, University of St Andrews, St Andrews, UK. [2]Laboratoire des Sciences du Climat et de l'Environnement (LSCE/IPSL), Saint Aubin, France. [3]School of Geography, University of Bristol, Bristol, UK. [4]School of Geography, Earth and Environmental Sciences, University of Birmingham, Birmingham, UK. [5]Department of Earth Sciences, Utrecht University, Utrecht, The Netherlands. [6]Department of Earth Sciences, University of Southern California, Los Angeles, USA. [7]Geological Survey of Norway, Trondheim, Norway. [8]Department of Geological Sciences, Stockholm University, Stockholm, Sweden. [9]School of Geography, Geology and the Environment, University of Leicester, Leicester, UK. [10]Department of Geology, Trinity College Dublin, The University of Dublin, Dublin, Ireland. ✉e-mail: molly.trudgill@lsce.ipsl.fr

Atlantic Magmatic Province (CAMP, e.g., ref. 4), which was extruded in several pulses spanning from the uppermost Triassic to the lowermost Jurassic[3,5]. However the exact nature of the global environmental changes that triggered the biotic crisis remains elusive[6,7]. Globally, carbon isotope stratigraphy shows a sharp negative carbon isotope excursion (CIE) between 1−8 ‰ coincident with the mass extinction event[4,8–12], termed the 'initial' CIE. This initial negative excursion is followed by a positive rebound and then another negative CIE, termed the 'main' CIE, around 100 kyr after the initial CIE (1–4 ‰, Fig. 1)[4,8,10–13]. The initial and main negative CIEs have been measured in both organic and inorganic substrates, in a suite of sections within and between basins from both hemispheres, and in varying sedimentary facies[4,8–12]. The initial CIE is thought to have been driven by the input of isotopically light carbon due to CAMP volcanism, but the exact mechanism is uncertain[4]. Previous studies have linked this CIE to volcanic $CO_2$ outgassing[4,14], with additional indirect contributions from contact metamorphism of carbon-rich host rocks, including methane release[4,10,14–18]. Reduced productivity due to the biotic crisis during the extinction event may have further contributed to the initial CIE[10]. Similar carbon sources are proposed for the main CIE[4,11,19]. The return to more positive values that follows the extinction is suggested to have been driven by the recovery of the organic carbon pump[9,14,19], in combination with enhanced burial of organic matter under

geographically expanded anoxic water column conditions initiated during and following the initial negative CIE[20,21].

Several kill mechanisms linked to CAMP volcanism have been proposed to explain the environmental and biotic perturbations during the Triassic−Jurassic transition. Elevated atmospheric $CO_2$ may have led to inhospitable temperatures[6] and is proposed to have led to the expansion of oxygen minimum zones as well as photic zone euxinia[6,20]. Conversely, $SO_2$ emissions are proposed to have led to short term cooling and acid rain[7], and other volatiles emitted due to CAMP activity, such as polyaromatic hydrocarbons and mercury, have been proposed to cause toxicity in plants[7]. Ocean acidification is also suggested as a kill mechanism based on several lines of indirect evidence at the end−Triassic mass extinction event[22]: marine calcifiers preferentially became extinct[23], replaced by agglutinated foraminifera and organic-walled disaster taxa[23,24], and carbonate sedimentation rates decreased worldwide[25]. Furthermore, multiple taxa, including carbonate producers, do not fully recover until later in the Jurassic, nearly 2 million years after the extinction event[26], indicating a prolonged perturbation of ecosystems beyond the initial CIE and associated extinction (e.g. ref. 26). However, these features can result from other ecological factors and provide no direct constraints on the scale of acidification. Reconstructions of ocean pH are thus required to test the potential influence of ocean acidification, carbonate

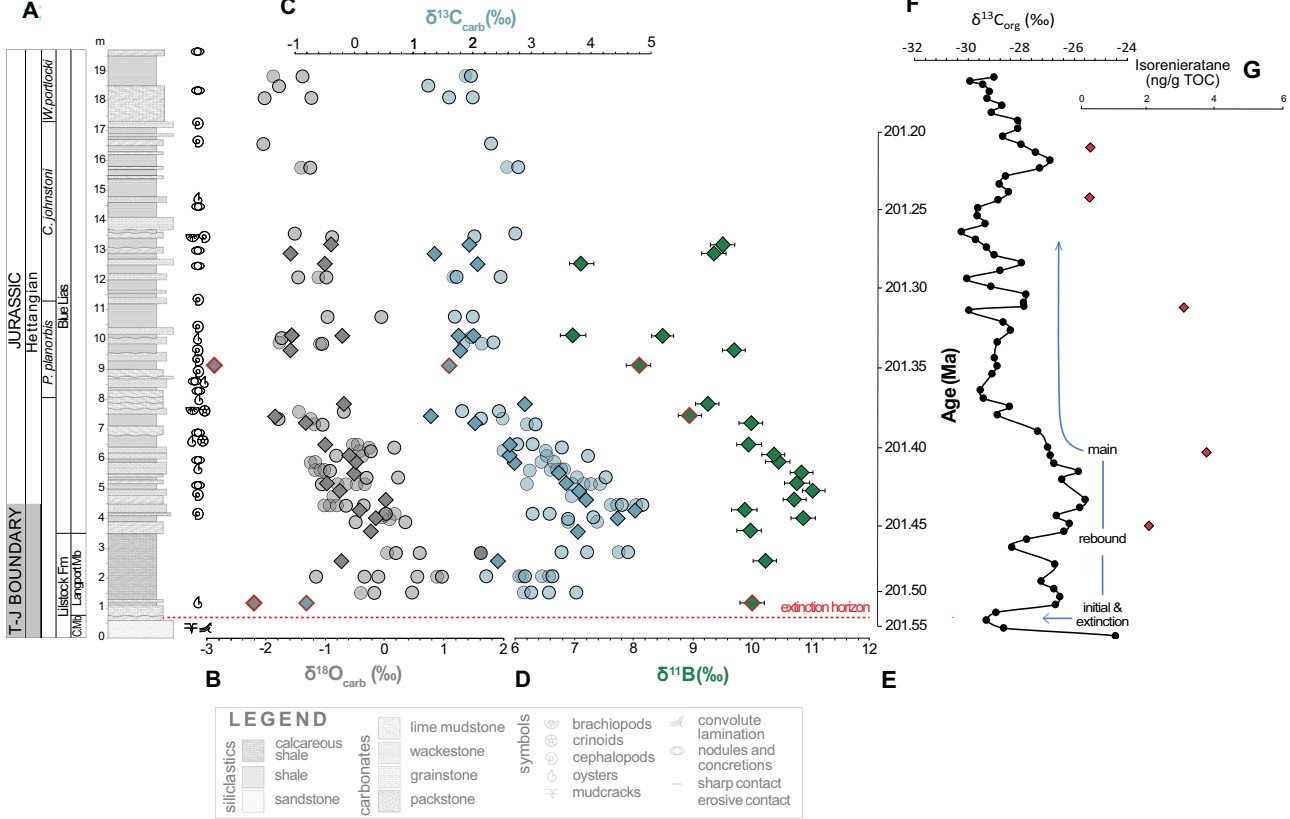

**Fig. 1 | Overview of lithological and geochemical data from Lavernock Point. A** Stratigraphy logged by the authors. (**B**) δ[18]O and (**C**) δ[13]C from this study (diamonds) and Korte et al.[19] (circles) (**D**) and δ[11]B (this study) measured in the fossil oyster *Liostrea hisingeri*. **E** Representative age scale after Xu et al.[13] from correlation with the St Audrie's Bay section. Rejected samples with δ[18]O and/or δ[13]C isotopically offset from the main trend or high Al/Ca are highlighted with a red border, though these samples show no notable difference in δ[11]B. **F** δ[13]C_org from St Audrie's Bay[4] using the age model from the relative astronomical time scale of Xu et al.[13] and the radiometric age of the initial carbon isotope excursion (CIE) from Blackburn et al.[3] rescaled to depth in this section (**G**) Isorenatine concentration from Jaraula et al.[44]

indicating prolonged euxinia at St Audrie's Bay following the extinction event. The records from Lavernock Point (**A−D**) cover the positive rebound and 'main' negative carbon isotope excursion. The position of the extinction horizon at Lavernock Point correlated with the 'initial' carbon isotope excursion is indicated by a red dashed line. Ammonite zones are from correlation with the section of Korte et al.[19]. Triassic−Jurassic boundary position is from Korte et al.[12], with uncertainty represented by grey shaded area. Boron isotope error bars represent long term external error based on repeated measurements of standards run with the same method in the STAiG lab (± 0.25 ‰).

undersaturation, and elevated $CO_2$ on environmental and biotic perturbations in this critical interval.

To address this knowledge gap, we analysed well-preserved fossilised oysters (*Liostrea hisingeri*; see SI/Methods) collected from Lavernock Point, Wales, for their boron isotope composition ($\delta^{11}B$), which is a proxy for ocean pH. Lavernock Point was located in an epicontinental sea at the Triassic–Jurassic boundary time interval (Fig. S1) and the sedimentary sucession is comprised largely of interbedded limestones, marls and mudstones (see Methods and Fig. 1A). Fossilised oysters from this locality have previously been shown to preserve a 2 ‰ negative carbon isotope excursion, corresponding to the main negative CIE at the end-Triassic, together with a drop in $\delta^{18}O$ caused by an increase in seawater temperature[19], and the stable isotope data from the oysters measured in this study largely follow these patterns (Fig. 1B, C).

## Results and discussion

Coincident with the $\delta^{13}C$ and $\delta^{18}O$ excursions, we find a drop in $\delta^{11}B$ of at least 3.3‰ lasting ~50 kyr (Fig. 1D). Even qualitatively, the presence of such a drop is indicative of a lowering of ocean pH and thus provides the most direct evidence to date for ocean acidification associated with the main CIE and carbon release at the Triassic–Jurassic boundary (Fig. 1). Beyond this qualitative interpretation, $\delta^{11}B$ in carbonate minerals can be translated into absolute pH values based on the known effects of pH on B speciation in seawater, the signature of which is incorporated into carbonates. However, conversion of $\delta^{11}B$ to absolute pH is hindered in deep time by the lack of vital effect calibrations for extinct species and poor constraints on the boron isotope composition of seawater ($\delta^{11}B_{SW}$) which may change on multi-million year timescales. We therefore take a conservative approach to translate the measured change in $\delta^{11}B$ into the minimum possible change in ocean pH.

### Minimum pH change

To provide a minimum estimate of pH change (Fig. S2B), we conservatively assume a one-to-one relationship between $\delta^{11}B$ in *L. hisingeri* and $\delta^{11}B$ of borate in seawater (which is a function of seawater pH). All existing calibration gradients are less than or equal to one[27], with a shallower gradient giving a larger change in $\delta^{11}B$ of borate (and thus pH) for the same measured $\delta^{11}B$ excursion. Although molluscs may upregulate their internal pH[28], this does not preclude sensitivity of their boron isotope composition to external pH in seawater, as seen in various other organisms, such as brachiopods and corals[29,30]. The oysters measured here do not have an obvious modern counterpart making it difficult to ground-truth any vital effects in the modern day, so to account for the uncertainty on species calibration we place minimal constraints on its gradient, allowing it to span the full range observed for modern calcifiers in the literature (Table S1). It is also necessary to determine the $\delta^{11}B_{SW}$ which results in minimum pH change. This calculation is done by establishing the maximum initial pH ($pH_i$) prior to the excursion, because a $\delta^{11}B$ excursion of a given size represents a larger pH change when $pH_i$ is low compared to when $pH_i$ is high (see Fig. S2A). We use a Latin hypercube random sampling strategy to determine the maximum $pH_i$ by taking the minimum possible atmospheric $CO_2$ level for the end–Triassic from independent reconstructions (500 ppm $\pm$ 50 ppm[31–33]), and the maximum ocean saturation state (10.7 $\pm$ 0.15[34]). Combining these constraints with estimates of calcium, magnesium, epsilon, salinity and temperature, where we again explore the extrema of their uncertainty which minimizes the change in pH we reconstruct (see table S1), yields a maximum $pH_i$ estimate of 8.18 at 95 % confidence. Using the average pre-excursion $\delta^{11}B$ measured from oysters (10.5 ‰ $\pm$ 0.036 ‰) we compute $\delta^{11}B_{SW}$ (31.1 ‰ $\pm$ 0.91 ‰). $\Delta pH$ is subsequently estimated by keeping all parameters the same (Table S1) except the measured change in $\delta^{11}B$ (–3.36 ‰ $\pm$ 0.06 ‰) and temperature (+ 3.0 °C $\pm$ 2.4 °C) reconstructed from $\delta^{18}O$ (Fig. 1). With these

assumptions, we calculate a minimum pH change of −0.29 units at 95 % confidence, from a $pH_i$ of 8.24 to a nadir of 7.93.

### Probabilistic pH and $CO_2$ change estimate

While this minimum pH change is robust, it is also unlikely to be representative of the true change in surface ocean pH which occurred over this event because it requires that all parameters were acting in concert to minimise pH change. Additionally, atmospheric $CO_2$ can also be calculated from estimates of ocean pH, provided that constraints can be placed on a secondary parameter within the ocean carbonate system (such as alkalinity or saturation state). We thus also calculate a suite of possible pH and $CO_2$ evolutions that are consistent with our $\delta^{11}B$ record (Fig. 2) and saturation state remaining above 1, consistent with continuous carbonate deposition at our site throughout our record, propagating full, conservative uncertainties on initial saturation state and atmospheric $CO_2$, and species calibration, calcium, magnesium, epsilon, salinity and temperature (rather than their extrema—see Table S1). In 95% of scenarios the drop in pH is >0.41 and the change in atmospheric $CO_2$ is >1300 ppm, equivalent to >1 doubling of $CO_2$. From this $\delta^{11}B$ record alone, we are unable to rule out much larger changes in pH (>1 unit) and atmospheric $CO_2$ (>10,000 ppm), owing primarily to the uncertainty in species calibration and uncertainties in saturation state and the evolution of alkalinity, respectively. This highlights the benefit of using the change in $\delta^{11}B$ to target minimum change in pH and atmospheric $CO_2$, which at present are constrained more robustly than absolute values. This record also allows us to better constrain a species calibration gradient for *Liostrea hisingeri*, which can be no lower than 0.8, and the initial $CO_2$ concentration, which according to our calculations is unlikely to be >3000 ppm. Below or above these limits the change in pH reconstructed is too great to allow the saturation state to remain above 1 throughout the record, as is consistent with continuous carbonate deposition throughout our record at this site.

We note that scatter in the record may result from the short life span of individual oysters, which record conditions representing ~10 yr time periods. Changes in the degree of scatter could represent changes in short term variability, for instance due to outgassing pulses, or environmental stress experienced during biomineralization, but given the small number of samples within a given interval, we avoid interpretation of this variability, and focus instead on the general trends, taking a 18 kyr smoothing window through the data (Fig. S3, see methods for discussion of smoothing interval). Previous estimates of atmospheric $CO_2$ during the Triassic–Jurassic carbon cycle perturbations come from fossil leaf stomata and pedogenic carbonate $\delta^{13}C$[32,33,35–37]. Reconstructions based on leaf stomatal densities are of low temporal resolution but record an increase of between 700 ppm and 1800 ppm across the end–Triassic mass extinction, beginning around or just before the initial negative CIE and returning to background levels after the Triassic–Jurassic boundary[32,33,37]; however, due to their low resolution, stomata-based reconstructions could miss two distinct atmospheric $CO_2$ increases associated with the initial and main negative CIEs. Further, some modern species have an upper limit above which they are no longer sensitive to $CO_2$; this varies between species but can complicate the use of this proxy in high $CO_2$ time periods like the Triassic–Jurassic[38]. Reconstructions from pedogenic carbonates indicate pulses of $CO_2$ release around 2000 ppm associated with the individual major CAMP flows, with a subsequent recovery between them[35,36]. The magnitude of the largest $CO_2$ change from pedogenic records, associated with the first CAMP volcanic unit and correlated with the extinction event, is between 2200 and 3000 ppm[35,36], larger than those from stomatal records, but within the range permitted by our data[37,38]. The second of these $CO_2$ pulses may also be temporally correlated with the main CIE and thus the pulse of ocean acidification we find here, though this is somewhat tentative given correlation uncertainties between sections used for pedogenic $CO_2$

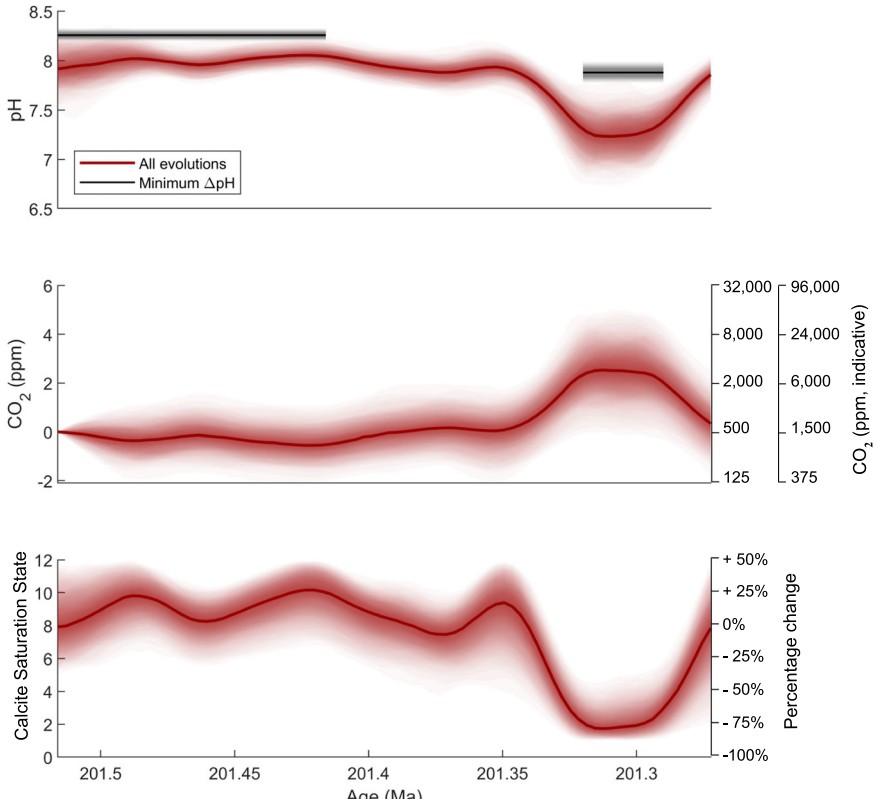

**Fig. 2 | pH, calcite saturation state and atmospheric CO₂ reconstructed from oyster δ¹¹B.** 95% confidence intervals are shaded for all simulated evolutions (red envelope), with median values shown by central line. Calculations used to generate these curves are described in the methods. Results obtained using the minimum ΔpH method are shown by the grey lines, with uncertainty represented by the grey shaded window. The minimum pH method take the average δ¹¹B values pre and during the excursion, with the length of the grey bars indicating the time periods averaged. Atmospheric CO₂ is plotted as doublings, which is relatively robust to uncertainties in initial conditions, with indicative CO₂ concentrations using the minimum and mean estimate of initial CO₂ generated by our calcualtions shown in the additional axes. Relative timescale using astrochronology of Xu et al.[13] by correlation with St Audrie's Bay, converted to absolute time using an age of 201.564 Ma for the initial negative carbon isotope excursion[3,15].

estimates and marine sections recording the main CIE[12,39]. The CO₂ estimates reconstructed here are thus within the range of other proxies for the Triassic–Jurassic carbon cycle perturbations, though we note they may not cover the exact same interval. Our data also provide robust evidence from co-located samples that the main negative CIE itself – as well as the initial CIE based on the previous reconstructions described above – was associated with an increase in atmospheric CO₂, suggesting a drop or hiatus in CO₂ rise during the positive carbon isotope rebound. This interpretation is consistent with the increased organic carbon burial thought to drive the positive δ¹³C rebound[9,14,19,21], which would have drawn down atmospheric CO₂ between the initial and main CIE.

## Carbon source

Constraining the magnitude of the CO₂ change associated with the main negative CIE allows determination of the isotopic composition of the carbon input, providing constraints on the predominant carbon source. Various carbon sources and processes have been proposed to have caused the Triassic–Jurassic carbon cycle perturbations and are each distinguished by their δ¹³C signature. CO₂ emissions from CAMP volcanism would have likely had δ¹³C values of −5 ‰ to −8 ‰[40], whereas carbon from biomass burning would have had a δ¹³C value of ~−22 ‰[41]. Depending on its lithology, the assimilation and contact metamorphism of CAMP intruded sedimentary host rock could have released carbon with δ¹³C values between 0 and −41 ‰[16], and/or triggered the release of thermogenic methane with a δ¹³C composition typically between −30 and −50 ‰ (e.g. ref. [16]). In addition, methane clathrates

are extremely ¹³C depleted, with δ¹³C lighter than −60 ‰[34]. With an independent constraint on the mass of carbon released (i.e., based on pH or CO₂ from our study), the magnitude of the associated CIE can be used to determine the likely δ¹³C of the carbon source.

We compare the main negative CIE measured in oysters from this section (this study and ref. [19]), using the difference between the average oyster δ¹³C value at the pre-event δ¹³C peak (3.62–4.31 m) to the average value in the CIE δ¹³C trough (7.17 – 12.89 m) (Fig. 1C). This gives a CIE magnitude of 2.2 ‰, which is within the 1–4 ‰ range of CIE magnitudes reconstructed worldwide[4,8,10–12], and provides the most robust value to pair with our δ¹¹B record from the same site. We use a generic cGENIE simulation set of carbon injections with δ¹³C characteristic of the carbon sources outlined above, including a carbon source with δ¹³C of −12 ‰ to represent a mixture between organic and volcanogenic carbon[1]. We test a range of onset durations for the main CIE, including the ~70 kyr duration constrained by the orbital timescale of Xu et al.[13], and the ~400 kyr radio-isotopic age estimate from Yager et al.[8]. The simulated Earth system response to the prescribed carbon injections accounts for the influence of major feedbacks, such as weathering and carbonate compensation.

Interpolating the model results to our best CIE estimate constrains the δ¹³C composition of the carbon source responsible for the pH excursion measured here (≥ 0.29 pH units) to be heavier than −12 ‰, ruling out a predominant source with an extremely isotopically depleted composition, such as methane, and indicating substantial mantle-derived carbon, likely alongside some organic component (Fig. 3). For a longer onset, the same pH excursion requires an even

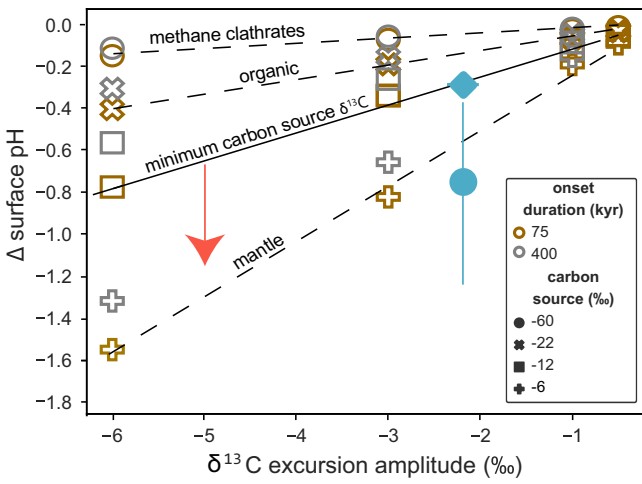

**Fig. 3 | Model and data derived constraints on carbon sources.** pH change modelled in cGENIE for different carbon sources and sizes of negative carbon isotope excursion (CIE) are shown in the symbols. Note that the simulation set by Vervoort et al. sampled a wide range of emission event configurations. The specific features of the Triassic–Jurassic emission events (duration, CIE, pH change) are not explicitly simulated but fall within the matrix of sampled generic emission events. Hence, we interpolate between the nearest members of the simulation result matrix to derive estimates for the main and initial CIEs. Different modelled onset durations are represented by different colours: 75 kyr (brown) or 400 kyr (grey). The minimum pH change we reconstruct (0.29 pH units) is shown by the blue diamond, with our estimate propagating full uncertainites shown by the blue circle, with the bar representing the upper and lower 95% confidence intervals. Our minimum estimate of pH change for the main CIE indicates that if the initial CIE, which has a magnitude of 5 ‰, was caused by the same source, it would have been associated with an even larger pH change (red arrow). Reconstructed onset durations for the initial and main CIE are -25–50 and 70–400 kyr respectively[8,13].

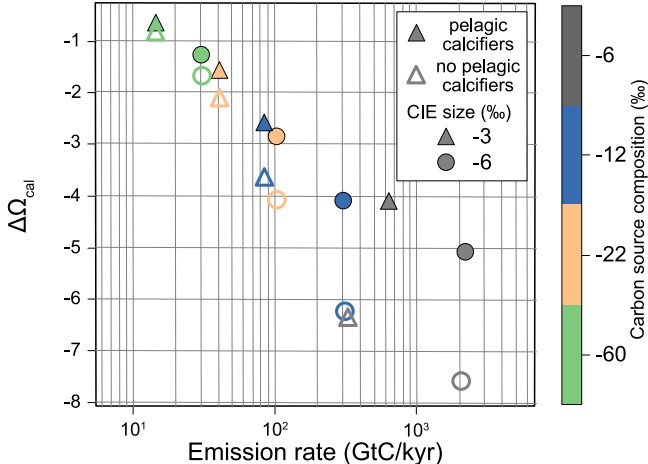

**Fig. 4 | Modelled saturation state decline.** Saturation state decline modelled in cGENIE for different carbon sources (indicated by symbol colour) and different sizes of negative carbon isotope excursion (CIE; symbol shape) for runs with a 75 kyr onset duration. With the same carbon source as the main CIE (>−12‰), we interpolate an emission rate of >500 GtC/kyr associated with the initial CIE (-5 ‰) and thus a >6 drop in calcite saturation state in an ocean without pelagic calcifiers. Open symbols represent model runs simulating conditions without pelagic calcifiers and closed symbols represent runs with pelagic calcifiers.

larger amount of carbon, implying an even heavier $\delta^{13}C$ composition and thus primarily mantle derived carbon source (Fig. 3). The magnitude of carbon input required to drive the drop in pH we reconstruct (>10,000 Pg C; Fig. S4) is well aligned with estimates of CAMP $CO_2$ emissions alongside contact metamorphism of the host rock[16,18,42,43]. One potential caveat is that enhanced burial of organic matter, which may have followed the extinction event[20,21], would have drawn down isotopically light C and, if carbon burial was further enhanced during the main CIE, would allow for a carbon source somewhat isotopically lighter than reconstructed here. A second caveat is that reverse weathering feedbacks could have further exacerbated acidification (see below), yet are not accounted for in current modelling. Nonetheless, given the relatively high $\delta^{13}C$ value constrained for the carbon source, it was likely largely mantle-derived and/or had a significant contribution from assimilation of carbonate host rock, with possible secondary inputs from a more isotopically depleted organic carbon source.

## Ocean acidification and undersaturation as a kill mechanism at the initial CIE

Given an initial surface ocean calcite saturation state of ~8, as is suggested prior to the diversification of pelagic calcifiers (5.0-10.7[34]), the minimum drop in pH estimated above would be associated with a decrease in saturation state of ~6 (Fig. 2). Both cyclostratigraphic and radiometric estimates suggest the initial negative CIE had a more rapid onset (-25–50 kyr[8,13]) and in most sections around the world it had a larger magnitude (5 ‰ at St Audrie's Bay[4], between 1 and 8 ‰ in other sections[8,10–12]). There are no oysters preserved across the initial CIE and bulk rock records do not preserve primary geochemical signals at this locality (see SI), so it is not

possible to extend the $\delta^{11}B$ record across the extinction horizon itself. However, if the initial CIE was caused by the same carbon source as the main negative CIE (an assumption we discuss below), its larger size (5‰[4]) and faster onset (-25–50 kyr[8,13]) would have meant it was associated with a greater carbon emission rate (Fig. 4), and thus a two times greater pH change than minimum estimates for the main CIE. Given the faster onset for the initial CIE, for an equivalent emission size it would have been associated with a roughly two fold increase in emission rate and thus a 1.5 times greater saturation state decline (Fig. 4).

The paired data and modelling indicate that the main CIE was predominantly driven by mantle-derived carbon, but additional carbon sources may have driven the initial negative CIE, such as methane released from destabilised ocean-floor clathrates or by thermal cracking of organic-rich sediments, or $CO_2$ released from other sediments through thermal metamorphism[4,10,14–16,18]. Carbon released from these alternative sources would have had a more $^{13}C$-depleted isotopic composition and would have caused a smaller pH drop for the same CIE. Nonetheless, the average surface ocean calcite and aragonite saturation state would still approach undersaturation if the carbon source had a $\delta^{13}C$ of −12 ‰ given a relatively rapid pace of carbon input (Fig. 4)[1]. This result suggests that even if the initial negative CIE had an almost 20 % contribution from methane or a 50 % contribution from sedimentary carbon, ocean acidification could still have driven undersaturation. A more severe ocean acidification event associated with the initial CIE than the main CIE is supported by the reduction in global carbonate sedimentation rates, paucity of preserved calcareous fossils, and preferential extinction of marine calcifiers associated with the initial CIE and extinction event but not the main CIE[23–25]. These results therefore suggest that ocean acidification, and associated undersaturation, was a likely kill mechanism during the end–Triassic mass extinction.

## Prolonged carbon cycle perturbations - biotic impacts and a role for reverse weathering?

The $CO_2$ system perturbation we reconstruct through the main CIE would have had profound biological and environmental consequences, despite being on timescales on which changes to saturation state are likely to be at least partially buffered[2]. The impact of

prolonged levels of elevated atmospheric $CO_2$ is evidenced by pronounced and continued global warming (Fig. 1), with coincident ocean anoxia and euxinia[44] (Figure 1G) indicating widespread ocean deoxygenation at this time, possibly explaining the prolonged biotic and ecological disturbance[26].

The recovery of ocean pH around 201.28 Ma is relatively rapid and coincides with the global emergence of silica dominated benthic ecosystems[26]. A potential explanation for these two observations is linked to the topical idea that reverse weathering – the uptake of cations and silica by authigenic clays on the seafloor – may have played an important role in major climatic transitions (e.g. [45,46]). We hypothesise that elevated $CO_2$ and global warming at the Triassic–Jurassic transition would have led to enhanced rates of silica delivery to the ocean via continental weathering, coupled with a reduced silica sink in the ocean due to suppression of silica producers in hot, low-oxygen conditions. The resulting build up of silica would have acted to promote reverse weathering reactions, which removed cations from seawater and lowered ocean alkalinity, acting as a feedback to prolong the high $CO_2$, low pH interval suggested by our data. The sudden emergence of silica producers at the tail end of the early Jurassic recovery would have curtailed this reverse weathering process, re-elevating ocean alkalinity and pH and lowering atmospheric $CO_2$. This mechanism may explain the rapid recovery of ocean pH at ~201.28 Ma coincident with the emergence of siliceous fossils and also provides an explanation for the decoupling of $\delta^{11}B$ and $\delta^{13}C$ (which would not have been noticeably impacted by the inorganic alkalinity cycling). Together with the growing body of evidence pointing to a role for reverse weathering in prolonging $CO_2$ highs[45,46], our data highlight the importance of better understanding this process and incorporating it into models which examine drawdown of anthropogenic $CO_2$, given its potential role in prolonging and exacerbating warming in the past.

**Mesozoic carbon cycle perturbations linked to background saturation state**

The data presented here provide further evidence of the pervasive link between major carbon cycle perturbations and mass extinctions in the early Mesozoic. While later Mesozoic and Cenozoic $CO_2$ injections drove notable environmental perturbations, the impact on calcifiers, while still detrimental, was less catastrophic. Prior to the dawn of pelagic calcifiers, the ocean would have maintained a higher background carbonate saturation state[34], and this would have led to a larger drop in the calcite saturation state for a given carbon injection (Fig. 4). Mechanistically, this inference results from the higher overall DIC content of the pre-pelagic calcifier ocean, meaning that for a given decrease in pH more carbon will be respeciated from carbonate ion to bicarbonate (Fig. S5). In addition, the lack of a sedimentary reservoir of pelagic carbonate can reduce the ocean's ability to buffer saturation state changes through the addition of alkalinity from carbonate dissolution, though this may be countered by the additional buffering capacity from the pre-pelagic calcifer ocean's overall higher alkalinity. Taken together, these factors mean that although the change in pH is similar for a given rate of carbon emissions, we model a ~1.5 times more extreme decrease in saturation state prior to the advent of pelagic calcifiers (Fig. 4, Figs. S6, S7).

The high background saturation state of the Mesozoic ocean thus means a given carbon cycle perturbation would have been associated with a greater decline in carbonate saturation state. Organisms which evolved and adapted to the high background saturation states of the Mesozoic oceans may also have found perturbations to lower saturation states more challenging to adapt to than faunas adapted to live in the modern ocean. Together with warm, low-oxygen conditions, exacerbated by palaeogeography[47], this would have contributed to the devastating consequences of Mesozoic ocean

acidification events, induced by major carbon emissions and associated global carbon cycle perturbations, which resulted in global mass extinctions at these times.

In summary, our data provide robust evidence for ocean acidification associated with carbon release at the Triassic–Jurassic boundary, showing a pH drop coincident with the main negative CIE. This prolonged interval of environmental perturbation seems likely to have contributed to continued biological stress. We find that the carbon source for this pH drop would likely have had a $\delta^{13}C$ value heavier than −12 ‰, suggesting it was predominantly mantle-derived and a result of CAMP volcanic activity. If carbon emissions from the same source also drove the larger and sharper initial negative CIE (100 kyr earlier) calcite and aragonite undersaturation would have been prevalent. Even if the initial negative CIE was partly a result of a more $^{13}C$-depleted carbon source, such as methane clathrates or sedimentary carbon from contact metamorphism of host rocks intruded by CAMP magmatism, significant ocean acidification and undersaturation would have likely still occured. Our conservative reconstructions of ocean pH change are at least as large as those observed for other extinction events in the Mesozoic (Permian–Triassic and Toarcian OAE, Fig. 5)[48,49]. These data present a clear basis for linking catastrophic ecosystem change with major carbon cycle perturbations, thus providing a strong indication that anthropogenic carbon emissions have the potential to significantly impact ocean chemistry and ecosystems: the comparable size of the observed changes in Triassic–Jurassic pH changes to reconstructions in response to IPCC high emissions scenarios (e.g. pH change of −0.43 by 2100 in RCP8.5[50]) stresses the importance of rapid action to reduce carbon emissions.

## Methods
### Geological and environmental setting
At the Triassic–Jurassic boundary interval, Lavernock Point consists of interbedded limestones, marls and mudstones deposited in an epicontinental seaway[51] (Fig. S1). The stratigraphic section logged and sampled comprises two major formations: the Lilstock Formation and the Blue Lias Formation. The Lilstock Formation is divided into the Cotham and Langport members. The Cotham Member is formed mostly of mudstone, siltstone and sandstone capped by an intensely deformed bed and large desiccation cracks, which have been interpreted as evidence for intense seismic activity and a hiatus in the section, respectively[52,53]. The initial CIE associated with the end–Triassic mass extinction event occurs at the top of the Cotham Member[4]. The overlying Langport Member is dominated by micritic limestones interbedded with mudstones. The Blue Lias Formation is comprised of alternating mudstones, limestones, marls and (organic-rich) shales and is further divided by several ammonite zones[13,51,54]. Prior to the first ammonite zone in the Blue Lias Formation are the Pre-*Planorbis* beds, which occur prior to the appearance of the first ammonite and are succeeded by the *Planorbis* zone, followed by the *Johnstoni* and *Portlocki* zones. The Triassic–Jurassic Boundary is marked by the first appearance of the ammonite *Psiloceras spelae*, which does not appear in any UK sections[55]. Thus, we use the position of the Triassic–Jurassic boundary from Korte et al.[12] for Lavernock Point which uses carbon isotope stratigraphy to correlate to the Kuhjoch section in Austria (the GSSP for the Triassic–Jurassic boundary) and uncertainty is represented with a grey shaded area on the stratigraphic section (Fig. 1).

At the Triassic–Jurassic boundary Lavernock Point was part of the European epicontinental seaway (Fig. S1). For some proxies this paleogeographic locality could be a limitation in recording global events; for example, greater temperature variability can occur compared to the global surface ocean average if a seaway is restricted, as it could be influenced by warming from land which is greater than that of open seawater. Despite this our temperature change, which may thus be influenced by a restricted seaway setting, and our CIE are similar to other global records (e.g. refs. 12,56), suggesting this location is

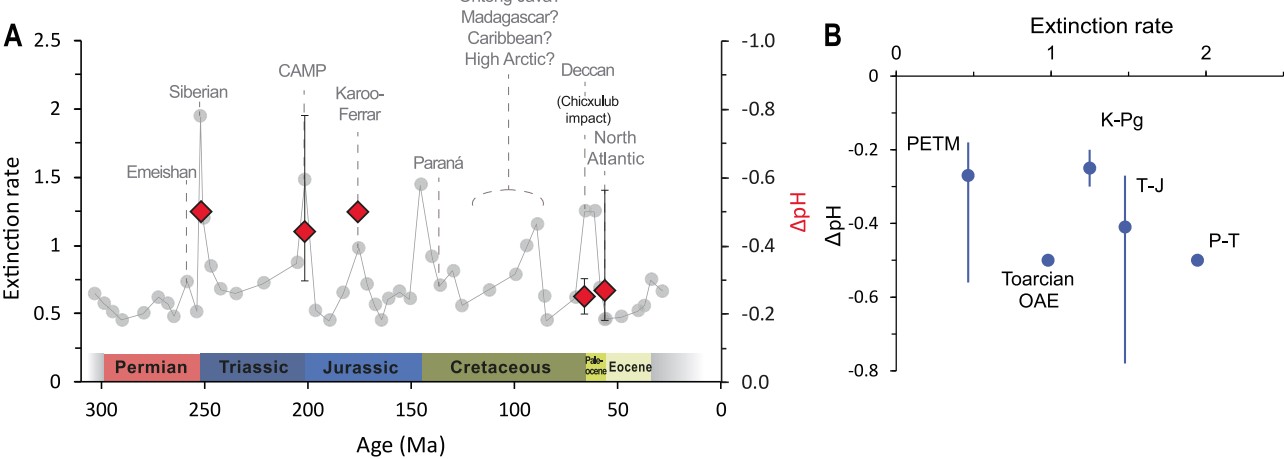

**Fig. 5 | Marine invertebrate extinction rate and pH excursion through time.**
**A** Marine invertebrate extinction rate (grey circles) and the large igneous provinces which correlate with major extinction events (labelled, after Clapham and Renne[78]). **B** Comparing extinction rate to reconstructed pH. Large igneous provinces (LIPs) are listed with question marks for Cretaceous Ocean Anoxic Events (OAEs) due to uncertainty surrounding which LIP corresponds to which OAE, compared to the magnitude of reconstructed pH decline (red symbols). Error bars are reported uncertainty for each pH estimate. From youngest to oldest, acidification has now been reconstructed during the Palaeocene-Eocene Thermal Maximum (PETM)[79], Cretaceous–Paleogene (K–Pg)[80], Toarcian OAE[48], Triassic–Jurassic (T–J, this study) and Permian–Triassic (P–T)[49] extinction events.

relatively representative of global conditions. Further, compared to temperature and $\delta^{13}C$, pH is likely to better track to the global mean[57]. Outside the high latitudes, pH varies by only -0.1 across the global preindustrial surface ocean, in part as the influences of temperature on seawater carbonate chemistry tend to dampen pH change[57]. This observation is also seen in our cGENIE experiments, where the maximum geographic range in pH is 0.1. Comparing pre-industrial to anthropogenic conditions, marginal seas do not show pH anomalies >0.05 pH units compared to nearby open ocean regions: the Mediterranean is -0.05 pH units lower than the open ocean; while high latitude seas and basins are -0.1 pH unit higher compared to the global mean, due to enhanced anthropogenic $CO_2$ uptake in colder waters, but are similar to adjacent high latitude open ocean sites[57]. Hence our pH reconstruction should, to first order, track global conditions.

## Sample collection and preparation
The stratigraphy at Lavernock Point was logged by the authors and samples of the oyster species *Liostrea hisingeri* were collected, which form their shells from low-Mg calcite (Fig. 1). This stratigraphy was correlated to the sedimentary log of Korte et al.[19], which in turn has been correlated to the section at St Audrie's Bay[12]. We use an age of 201.56 ± 0.22 Ma (U-Pb CA-TIMS on zircon[3]) for the initial negative CIE and then use the relative astronomical timescale of Xu et al.[13] from St Audrie's Bay to determine the sedimentation rate.

Samples were collected from micrite layers. The surface was chipped away to extract well-preserved embedded oysters with no sign of recrystallisation. Preservation is illustrated by thin sections and SEM images which show original shell textures and foliations, suggesting excellent preservation and no meteoric diagenesis (Fig. S8). Oyster fragments were assessed optically using a microscope and any with dark inclusions, yellow discolouration or dissolution textures were removed. When measured, less optically pristine material yielded $\delta^{11}B$ largely within error of pristine material (Fig. S9), but as this may not always be the case, only the best preserved material was taken for analysis. Well-preserved flakes were crushed to -0.5 mm average particle size using an agate pestle and mortar for cleaning and analysis.

## Oxygen and carbon isotope analysis
Carbon and oxygen stable-isotope analyses were conducted in the Geolab at the Department of Earth Sciences at Utrecht University on a Thermo Fisher Scientific MAT 253 and MAT 253 plus isotope ratio mass spectrometer (IRMS) coupled to a Kiel IV carbonate device. Stable isotope values are reported versus V-PDB by analysing carbonate standards ETH-1, ETH-2 and ETH-3[58] during each automated run. As an additional check standard IAEA-C2 is measured to monitor long-term reproducibility. The $\delta^{13}C$ and $\delta^{18}O$ values of IAEA-C2 showed an external reproducibility (1 standard deviation) of 0.05‰ and 0.12‰, respectively.

These analyses (and additional trace element measurements described below) allow us to screen samples for diagenesis as altered samples are likely to be driven isotopically light. Two samples appear offset isotopically light relative to the overall trend for $\delta^{18}O$ and/or $\delta^{13}C$ (highlighted in red on Fig. 1) and one sample has anomylously high Al/Ca, so they are not included in further calculations.

The $\delta^{13}C$ and $\delta^{18}O$ excursions discussed in the main text are calculated from our data by taking the difference between the average pre-event oyster $\delta^{13}C/\delta^{18}O$ composition and the average $\delta^{13}C/\delta^{18}O$ value in the trough.

## Boron isotope analysis
Crushed samples were cleaned following the methods of Rae et al.[59] and dissolved in 200 μl B-free MilliQ and 50−125 μl 0.5 M $HNO_3$. Briefly this involved clay removal by sequential MilliQ washes, an oxidative clean designed to remove organics followed by a weak acid (0.0005 M $HNO_3$) leach and dissolution.

Samples were analysed for trace element composition by inductively-coupled mass spectrometry (ICPMS) on an Agilent 7500 at the University of St Andrews. Samples with high Al/Ca (>100 μmol/mol), which would be indicative of clay contamination, or high Mn/Ca or low Sr/Ca, which can indicate diagenesis, were not offset from the overall trend, though one sample with anomylously high Al/Ca was removed from the dataset for further calculations (highlighted in red on Fig. 1, Fig. S10). Boron was then purified from the sample matrix by separation using a boron specific ion-exchange resin, Amberlite IRA-743[60], following the column chemistry procedure of Foster[61]. Boron isotope composition was then analysed on the Neptune Plus multi-collector ICPMS (MC-ICPMS) at the University of St Andrews following the procedures of Rae[62], using 0.3 M HF in samples, standards and blanks to aid washout[63]. Boric acid standards and the carbonate standard JCP-1, run alongside these samples and long term in this lab, are

within analytical uncertainty of their published values (run alongside samples: AE121: 19.63, 2 SD 0.13, $n = 4$; BIGD: 14.71, 2 SD 0.23, $n = 4$; JCP-1: 24.19, 2 SD 0.68, $n = 2$; published values: AE121: 19.63, 2 SD 0.15, $n = 504$[64]; BIGD: 14.77, 2 SD 0.19, $n = 209$[64]; JCP-1: 24.25, 2 SD 0.22, $n = 103$[65]).

We note that scatter in the record may result from the short life span of individual oysters, which record conditions representing ~10 yr time periods. Changes in the degree of scatter could represent changes in short term variability, for instance due to outgassing pulses, or environmental stress experienced during biomineralization, but given the small number of samples within a given interval, we avoid interpretation of this variability, and focus instead on the general trends.

### Reconstructing temperature

We use $\delta^{18}O$ to estimate temperature, which requires three components: measured $\delta^{18}O$, $\delta^{18}O_{SW}$, and a calibration of $\delta^{18}O$ and $\delta^{18}O_{SW}$ to temperature. We generated paired measurements of $\delta^{18}O$ for all samples. In two instances the signal appears isotopically light relative to the overall trend, which we attribute to potential diagenetic alteration. These two measured $\delta^{18}O$'s are replaced with estimates from a Gaussian Process conditioned on unaltered samples. $\delta^{18}O_{SW}$ is estimated using the average $\delta^{18}O_{SW}$ from nearby clumped isotope data by Petryshyn et al.[66] ($1.05 \pm 0.46$ ‰, 2 SD). We propagate uncertainty on the average $\delta^{18}O_{SW}$ into our temperature estimate. We trial four potential $\delta^{18}O$-temperature calibrations[67–70]. All four calibrations have nearly identical sensitivity to changes in measured $\delta^{18}O$, therefore the calculated change in temperature is nearly independent of the chosen calibration, but the absolute value is offset between different calibrations. We propagate uncertainty using two components (uncertainty in the absolute temperature, and uncertainty in the change in temperature) by using the youngest temperature estimate as a tiepoint.

### Reconstructing pH and atmospheric $CO_2$

The method used here to determine minimum pH change consistent with a measured $\delta^{11}B$ change (as outlined in the main text) is to first use independent constraints on the carbonate system to establish maximum initial pH (see Fig. S2). A $\delta^{11}B$ excursion of 3.36 ‰ ($\pm 0.06$ ‰) is used, calculated using a random sampling (Monte Carlo) process to take a median through the first 9 and lowest 2 data points, accounting for analytical uncertainty, and taking the difference and standard deviation of these groupings. Maximum initial pH is combined with measured pre-perturbation $\delta^{11}B$ to calculate $\delta^{11}B_{SW}$, assuming that measured $\delta^{11}B$ is representative of borate $\delta^{11}B$. Boron has a ~10 Myr residence time in seawater[71], so $\delta^{11}B_{SW}$ is assumed to remain constant on this timescale (<300 kyr) and is used with the minimum measured $\delta^{11}B$ to determine minimum recorded pH. The difference between maximum initial pH and minimum pH is the minimum pH change. Additional parameters are required to perform this calculation, including the boron isotope fractionation factor (epsilon) and the boric acid dissociation constant, $pK_B$ (which requires estimates of temperature, pressure, and salinity, and seawater calcium and magnesium concentrations). The values of required parameters are specified in Table S1. Uncertainties are propagated for these calculations using a Latin hypercube random sampling strategy. For each individual calculation of pH change in our ensemble, $\delta^{11}B_{SW}$, calcium, magnesium, epsilon, salinity, and pressure are held fixed (as these are invariant on this timescale) and only change with each iteration, while $\delta^{11}B$ and temperature are allowed to vary within their measured uncertainties. The sensitivity of our estimate of pH change to these parameters is show in Fig. S11.

Identifying the full range of possible pH evolutions follows much the same procedure, but both $\delta^{11}B$ and temperature are interpolated using a Gaussian process, and other input parameters have updated ranges as shown in Table S1. $\delta^{11}B$ is smoothed with an 18 kyr window. With a 50 kyr smoothing window, while the shape of the record

changes, the values we reconstruct are within error (e.g. median pH change 0.02 pH units smaller, Fig. S3), suggesting the values we reconstruct are robust to various interpretations of the scatter within the record. ANOVA analysis on data before and during the excursion shows a statistically significant difference, even when considering a wide interval as the excursion ($p = 0.0002$, Fig. S3).

As we also want to understand changes in atmospheric $CO_2$ concentration, it is necessary to quantify a second carbonate system parameter. The carbonate system has two degrees of freedom, so from any two parameters (e.g. pH and alkalinity) it is possible to calculate the rest (e.g. saturation state, DIC, atmospheric $CO_2$). From our records we reconstruct ocean pH, so by using constraints on any one of the other parameters we may reconstruct the rest of the carbonate system. Here we use alkalinity, as it is the parameter with the longest residence time so likely to be the most stable through our record. We quantify initial alkalinity using the parameters we have the best constraints on for the Triassic–Jurassic: initial $CO_2$ and saturation state, for which we take flat distributions consistent with broad ranges from proxy[31,33,35,36] or modelled[34] estimates for this time period. However, alkalinity is likely to have changed over the 300 kyr interval our samples span (present day residence time is roughly 100 kyr). To propagate the uncertainty in how alkalinity may have changed, we use a Gaussian process, conditioned on output from the cGENIE model, which shows the trend in alkalinity over a simulated $CO_2$ release. This produces alkalinity curves with an appropriate shape, which are then scaled so that the change is on the order of 100's to 1000's of micromoles/kilogram and allowed to be either positive or negative, reflecting the possibility for a strong weathering feedback or a strong reverse weathering feedback during the carbon cycle perturbation. From the continuous carbonate deposition at our site we know that saturation state never dropped below 1, allowing us to reject any scenarios where this occurs. Alongside reconstructions of atmospheric $CO_2$, this approach also calculates the change in calcite saturation state associated with our pH excursion.

Additional complexity arises where there are multiple constraints on individual parameters. Specifically, there are two constraints on $\delta^{11}B_{SW}$ (measured $\delta^{11}B$ + epsilon, and measured $\delta^{11}B$ + initial pH), and two constraints on the carbonate system (saturation state, and rate of change of alkalinity). As the technique described above is reliant on random sampling, it is possible to draw samples which are not consistent with the imposed constraints (e.g. when generating a $\delta^{11}B_{SW}$ using initial $\delta^{11}B_{borate}$ and initial pH, it is possible to calculate a $\delta^{11}B_{SW}$ incompatible with $\delta^{11}B_{borate}$ measured later in the record). When this happens, the samples are rejected, so that we obtain pH and $CO_2$ evolutions which do not violate any of the imposed constraints. This provides some narrowing of constraints on initial $CO_2$ values, as mentioned in the main text.

### Model simulation results

We compared our data to generic carbon injection simulations with the Earth system model cGENIE, specifically an extended version of the experiment set by Vervoort et al.[1] and a series of additional model experiments based on these. This includes an ensemble of carbon injection experiments with the Earth system model cGENIE. cGENIE contains a dynamic, isotope-enabled carbon cycle, as well as a range of other biogeochemical cycles and modules for the representation of ocean dynamics, sea-ice coverage, atmospheric energy and moisture balance[72]. The generic model configuration used by Vervoort et al.[1] provides a first-order estimate of the Earth system response to carbon injections during greenhouse climate conditions, including changes in marine carbonate burial, sea-air gas exchange and continental weathering[73,74]. This set up has simplified continental configuration and topography, 850 ppm initial $xCO_2$, 7.7 mean surface ocean pH, and 5.5 and 2.5 mean calcite and aragonite saturation states respectively, and is designed to approximate general Mesozoic and Cenozoic

greenhouse conditions. In the experiments of Vervoort et al.[1], cGENIE was forced to reproduce a range of negative $\delta^{13}C$ excursions in marine dissolved inorganic carbon to produce a data set that can be used to constrain net carbon emissions and their biogeochemical consequences from the duration, shape and size of a given marine $\delta^{13}C$ excursion. Vervoort et al.[1] also present examples of how their model ensemble can be applied to study $\delta^{13}C$ excursions in the geologic record. To cover the full range of suggested onset durations for the main CIE we added simulations of 400 kyr long CIE onsets, and a series of simulations with a higher initial saturation state/without a pelagic carbonate sink to better simulate early Mesozoic conditions prior to the Mid-Mesozoic pelagic calcifier revolution[34]. Specifically, we reduced pelagic carbonate production by 90% by changing the fixed PIC:POC ratio of export production. This results in a complete stop in pelagic carbonate burial. We partially compensate for the lack of pelagic carbonate burial by turning on saturation state-dependent carbonate deposition on reefs. We stabilize the marine carbonate system at a higher-than-pre-industrial mean surface calcite saturation state of 7.9 by reducing the overall weathering input. Mean surface $\delta^{13}C$ and organic export production remain unchanged compared to the Vervoort et al. default model setup. Global carbon fluxes in the Vervoort et al. default and new neritic model steady state are shown in Table S2. We use this extended model ensemble in combination with the observed $\delta^{13}C$ excursions and the reconstructed pH change to constrain net carbon additions and saturation state changes across the Triassic–Jurassic boundary interval (Fig. 3), to investigate the impact of emission rates and the presence or absence of pelagic calcifiers on saturation state decline (Fig. 4), and the total emissions associated with different pH excursions (Fig. S4).

## Data availability
The boron, carbon and oxygen isotope and trace element data generated in this study have been deposited in the PANGAEA database[75] [https://doi.org/10.1594/PANGAEA.983346].

## Code availability
Carbonate chemistry calculations are performed in Matlab, using the package available at https://github.com/St-Andrews-Isotope-Geochemistry/BuCC, which is validated against csys for the use case presented here. Code for the analyses and plots shown here is archived on Zenodo[76] [https://zenodo.org/records/15475089], with the raw data output found on Figshare[77] [https://doi.org/10.6084/m9.figshare.29094518.v1]. The specific version of the cGENIE.muffin model used in this paper is tagged as release v0.9.6 and has been assigned a (https://doi.org/10.5281/zenodo.3338584). The code is hosted on GitHub and can be obtained by cloning: https://github.com/derpycode/cgenie.muffin, changing directory to cgenie.muffin, and then checking out the specific release: $ git checkout v0.9.6. Configuration files for the specific experiments presented in this paper can be found in the directory: genie-userconfigs\MS\vervoortetal.2019. Details of the experiments, plus the command line needed to run each one, are given in the README.txt file in that directory. All other configuration files and boundary conditions are provided as part of the release. A manual detailing code installation, basic model configuration, plus an extensive series of tutorials covering various aspects of muffin capability, experimental design, and results output and processing, is provided on GitHub. The LaTeX source of the manual, along with a pre-built PDF file can be obtained, by cloning: https://github.com/derpycode/muffindoc. Model output files of this study are available from the corresponding author upon request.

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

## Acknowledgements

This work was supported by European Research Council Horizon 2020 research and innovation programme grant agreement No. 805246 to JWBR, NERC Large Grant NE/P01903X/1 to SEG and MA, NERC Independent Research Fellowship NE/L011050/1 to SEG, NERC IAPETUS PhD Studentships NE/RO12253/1 to MT, JCB, NERC GW4+ PhD Studentship S136361 to MA and NERC CENTA PhD Studentship NE/L002493/1 to DD. MT acknowledges the support of the EUR IPSL-Climate Graduate School (reference ANR-11-IDEX-0004-17-EURE-0006).

## Author contributions

J.W.B.R., S.E.G. and A.J.W. conceptualised the idea. Sample collection and fieldwork was done by J.W.B.R., M.V.M., F.C., D.D., C.V.R., D.S. and S.E.G. Data collection was done by J.C.B., M.V.M., M.C., W.G., R.G., W.L.H., A.M., N.N., M.M.F.R.S. and M.Z. and data analysis by M.T., J.W.B.R. and R.W. Modelling was done by M.A. and S.E.G. Supervision was done by J.W.B.R., A.B., C.V.R., R.C.J.S. and S.E.G. The original draft was written by M.T. with review and editing by M.T., J.W.B.R., J.C.B., R.W., M.V.M., M.A., A.B., D.D., W.L.H., A.L., A.M., N.N., C.V.R., M.R., D.S., R.C.J.S., ES, A.J.W., M.Z. and S.E.G.

## Competing interests

The authors declare no competing interests.
