## [Transparent Peer Review file · Nature Communications]

Pulses of ocean acidification at the Triassic–Jurassic boundary

Corresponding Author: Dr Molly Trudgill

Version 0:

Reviewer comments:

Reviewer #1

(Remarks to the Author)

(Remarks on code availability)

We thank the reviewer and editor for their positive remarks. In response to this review and the comments from the editor we have performed ANOVA analysis on our data, showing that the data in our isotope excursion is statistically significantly different from that prior to the excursion. We have moved some of the calculation methods to the main text to improve readability and addressed the other comments as detailed below.

This is a review of the article “Pulses of ocean acidification at the Triassic–Jurassic boundary”. The authors present boron isotope data from well preserved fossil oysters to reconstruct pH change across the Triassic Jurassic boundary. Earth system modeling was used to reproduce both carbon isotope excursion and estimated pH change, to provide insight to the source of carbon. The authors determine a >0.29 unit pH change, and suggest this as a primary driver of the extinction. Carbon is determined to be mantle derived with a magnitude of >10,000 GtC. Overall, this is a really neat dataset and comprehensive study and I highly commend the authors for this. I think that the authors have dealt with the 4 previous reviewer questions comprehensively. I have some minor notes below. I strongly recommend publication.

In reference to Reviewer 3 comments

Data inclusion – Here, I think it is ok to include / exclude the data as long as it is explicitly detailed or differentiated in the main text or supplement. I think the authors have dealt with this.

Signal and scatter – in two or three bins, a statistical analysis (e.g., ANOVA) would demonstrate this.

Thanks, we have performed this analysis showing the difference between points prior to and during our d11B excursion are statistically significant, even when considering a wider interval scatter during the excursion and included this in the methods and figure caption (lines 764-766):

ANOVA analysis on data pre and during the excursion shows a statistically significant difference, even when considering a wide interval as the excursion ($p = 0.0002$, Fig. S3).

Fig. S3 Data with standard 18 kyr smoothing window used in main text/Figure 2 (red) and a smoother 50 kyr fit (blue). With a smoother fit more smoothed dataset although the shape of the record changes, the magnitude of change remains similar. ANOVA analysis of data prior to (black points) and during the excursion (brown points) shows a statistically significant difference ($p = 0.0002$).

Other comments

Figure 2

- (B) should the vertical axis say (x background)
- (C) should the axis specify saturation of ...
- it could be nice to indicate on the figure what the grey bar means, readers glancing often just look at figures and not always the text.
- is there a corresponding grey bar for CO₂ and saturation state?

Thanks, have made these edits. There is no corresponding bar for CO₂ and saturation state because our minimum pH method only reconstructs pH.

Figure 5

- A cross plot of extinction rate versus pH change in the corner could be interesting.

Yes, have included as a panel on this figure

Line 165: Stomata index – are there lower/upper limits to this index?

Have included in the text (line 171-173):

Further, some modern species have an upper limit above which they are no longer sensitive to CO₂, this varies between species but can complicate the use of this proxy in high CO₂ time periods like the Triassic-Jurassic³⁸.

Carbon source - paragraph could be broken up a little?

Yes, have broken into three paragraphs.

Paragraph Line 267 on: framework from Isson et al. (2022), building on ref 47 as opposed to ref 46.

Have included this reference.

Extended Data figure 4 – capitalize Concentration on the vertical axis.

Done

Extended Data figure 12 – the horizontal axis doesn't match the length of the result. Also 'used in main text' – maybe specify the figures. "With a more smooth dataset..." sounds like a different dataset is used. Do you mean a smoother fit/curve.

Done

Isson, T. T., Zhang, S., Lau, K. V., Rauzi, S., Tosca, N. J., Penman, D. E., & Planavsky, N. J. (2022).

Marine siliceous ecosystem decline led to sustained anomalous Early Triassic warmth. Nature communications, 13(1), 1-1

This is a review of the article “Pulses of ocean acidification at the Triassic–Jurassic boundary”. The authors present boron isotope data from well preserved fossil oysters to reconstruct pH change across the Triassic Jurassic boundary. Earth system modeling was used to reproduce both carbon isotope excursion and estimated pH change, to provide insight to the source of carbon. The authors determine a >0.29 unit pH change, and suggest this as a primary driver of the extinction. Carbon is determined to be mantle derived with a magnitude of >10,000 GtC. Overall, this is a really neat dataset and comprehensive study and I highly commend the authors for this. I think that the authors have dealt with the 4 previous reviewer questions comprehensively. I have some minor notes below. I strongly recommend publication.

In reference to Reviewer 3 comments

Data inclusion – Here, I think it is ok to include / exclude the data as long as it is explicitly detailed or differentiated in the main text or supplement. I think the authors have dealt with this.

Signal and scatter – in two or three bins, a statistical analysis (e.g., ANOVA) would demonstrate this.

Other comments

Figure 2

- (B) should the vertical axis say (x background)
- (C) should the axis specify saturation of ...
- it could be nice to indicate on the figure what the grey bar means, readers glancing often just look at figures and not always the text.
- is there a corresponding grey bar for CO₂ and saturation state?

Figure 5

- A cross plot of extinction rate versus pH change in the corner could be interesting.

Line 165: Stomata index – are there lower/upper limits to this index?

Carbon source - paragraph could be broken up a little?

Paragraph Line 267 on: framework from Isson et al. (2022), building on ref 47 as opposed to ref 46.

Extended Data figure 4 – capitalize Concentration on the vertical axis.

Extended Data figure 12 – the horizontal axis doesn’t match the length of the result. Also ‘used in main text’ – maybe specify the figures. “With a more smooth dataset...” sounds like a different dataset is used. Do you mean a smoother fit/curve.

Isson, T. T., Zhang, S., Lau, K. V., Rauzi, S., Tosca, N. J., Penman, D. E., & Planavsky, N. J. (2022). Marine siliceous ecosystem decline led to sustained anomalous Early Triassic warmth. *Nature communications*, 13(1), 1-12.